# White-nose syndrome, winter duration, and pre-hibernation climate impact abundance of reproductive female bats

Sarah K. Krueger[1¤], Sarah C. Williams[2], Joy M. O'Keefe[3], Gene A. Zirkle[2], Catherine G. Haase[1] *

1 Department of Biology, Austin Peay State University, Clarksville, Tennessee, United States of America,
2 Environmental Division, US Army Fort Campbell, Fort Campbell, Kentucky, United States of America,
3 Department of Natural Resources and Environmental Sciences, University of Illinois, Urbana, Illinois, United States of America

¤ Current address: Environmental Resources Management, Seattle, Washington, United States of America
* haasec@apsu.edu

**Data Availability Statement:** We provide some of the published data in the Supplementary Materials, but other data not published cannot be shared with public access. We noted the owners of the dataset

## Abstract

White-nose syndrome (WNS) is an infectious disease that disrupts hibernation in bats, leading to premature exhaustion of fat stores. Though we know WNS does impact reproduction in hibernating female bats, we are unsure how these impacts are exacerbated by local climate factors. We compiled data from four southeastern U.S. states and used generalized linear mixed effects models to compare effects of WNS, pre-hibernation climate variables, and winter duration on the number of reproductive females in species across the range of WNS susceptibility. We predicted we would see a decline in the number of reproductive females in WNS-susceptible species, with the effect exaggerated by longer winter durations and pre-hibernation climate variables that lead to reductions in foraging. We found that the number of reproductive females in WNS-susceptible species was positively correlated with pre-hibernation local climate conditions conducive to foraging; however, WNS-susceptible species experienced an overall decline with the presence of WNS and as winter duration increased. Our long-term dataset provides evidence that pre-hibernation climate, specifically favorable summer weather conditions for foraging, greatly influences the reproduction, regardless of WNS status.

## Introduction

In sexually reproducing species, timing parturition to coincide with peak food resource availability ensures adequate energetic resources for raising offspring [1, 2]. Many mammalian species have evolved strategies to delay reproduction to wait out unfavorable conditions for sustaining newborns, such as lack of food, to ensure the success of their young [3]. For example, the African buffalo (*Synerus caffer*) synchronizes birthing with periods when protein content is abundant in forage. Researchers believe this strategy is triggered by environmental cues such as rainfall and sprouting of seedlings and saplings [4]. In another example, little brown

in our manuscript so readers may request data from these contacts. Restriction on sharing the data publicly is due to data being owned by a third-party organization. Please contact Fort Campbell Fish & Wildlife (5290 8th St, Fort Campbell, TN 42223; 931-472-5820) for data requests.

**Funding:** Funding for this research and support for S. Krueger was provided by an Intergovernmental Support Agreement with the U.S. Army (Order # CAMP-IGSA-20-02). The funders had no role in study design, data collection and analysis, decision to publish, or preparation of the manuscript.

**Competing interests:** The authors have declared that no competing interests exist.

bats (*Myotis lucifugus*) mate before hibernation, during peak fitness, and females store sperm for parturition post-hibernation emergence [5]. These reproductive delays are driven by the physiology of the mother and include delayed fertilization through sperm storage [6], delayed implantation of the zygote [7], and delayed development of young [5].

Delayed reproduction also permits trade-offs between reproduction and other elements of a species' life cycle, for instance, hibernation [3]. Reproductive delays throughout hibernation is a common strategy of many temperate insectivorous bat species in the family Vespertilionidae. Hibernation is a vital period in the life cycle of many mammals that require energy-saving strategies to maximize fitness over periods of food scarcity, such as during winter [8]. Hibernation is characterized by periods of torpor and arousal, where torpor is the short-term reduction in body temperature and metabolism that is interrupted by periodic arousals to normothermic body temperatures and metabolic rates [8]. Female bats delay fertilization or implantation during hibernation to minimize the energetic costs during winter while maximizing food availability for parturition [9]. Pups are born when prey availability is optimal and there is enough time to reach maximum adult mass before hibernation the following winter [6].

The phenology of emergence from hibernation by females is influenced by a balance between two contrasting factors: the benefit of giving birth early in the season and the potential risk associated with facing unfavorable spring weather or a shortage of food resources [10–12]. Likewise, juvenile survival is linked to the birth date of pups, with those born earlier in the summer having a greater likelihood of surviving their first year than those born later in the summer [10]. Thus the severity of winter and the duration of hibernation can drastically impact juvenile survival and potential population growth. For example, Burles et al. [13] found that *Myotis lucifugus* experienced delays in reproductive timing when their study area was experiencing unusually cool, wet weather from El Niño (in 1999). Low ambient temperatures increase the energetic costs of maintaining a normal body temperature and may result in prolonged gestation and lower reproductive success [13].

Though much research has determined the impacts of food availability and climate on delayed reproduction [10, 13–16], little is known about how the energetic costs of disease may alter this response. For example, white-nose syndrome (WNS), a disease of North American hibernating bats, increases energetic costs during hibernation through disruption of the torpor-arousal cycle, leading to loss of fat stores and high rates of mortality [17–19]. Physiological changes such as decreased fat mass, elevated $CO_2$ levels in the blood, dehydration, and increased fat consumption have been documented in WNS-affected bats [20]. These physiological changes can result in respiratory acidosis, hyperkalemia, and evaporative water loss, possibly stimulating increased arousal frequency that results in premature fat consumption [20, 21]. Due to the increased energetic requirements during hibernation in WNS-affected bats, fertilization is often postponed or completely forgone by mated females [21, 22]. Postponing fertilization to a later date can be problematic to WNS-susceptible species if juveniles lack adequate time to gain fat stores before hibernation the following winter [23].

Since its detection in New York in 2006 [24], WNS has been confirmed in at least twelve bat species in over half of the U.S. states and Canadian provinces (whitenosesyndrome.org); additionally, the causative fungus, *Pseudogymnoascus destructans*, has been found on six more species. Observed differences in fungal loads among species and resulting morbidity and mortality suggest variation in susceptibility to WNS across North American bats. Though the true mechanisms that drive this variation is unknown, various factors have been linked with interspecific variation in fungal loads, including sociality and grooming during hibernation, hibernaculum microclimate, skin microbiome, and the amount of pre-hibernation body fat [18, 25–29]. For example, the small-bodied tricolored bat (*Perimyotis subflavus*) hibernates in caves that sustain microclimates for high fungal growth, resulting in extremely high fungal loads and

prevalence rates (2.5–15.8˚ C; [18, 25, 28, 30, 31]). Other species, such as *Eptesicus fuscus* (big brown bat), persist with low fungal loads and are resistant to mass mortality; the mechanism behind these low loads is still unclear, but the skin microbiome has been hypothesized to support resistance [32, 33]. Across hibernating bat species in North America, differing hibernation behaviors and immune responses can result in variation in susceptibility to this disease.

Given the variation in vulnerability to WNS, it is crucial to evaluate how hibernation behavior, local climate conditions, and the duration of winter collectively influence the likelihood of survival, and in turn, reproduction. In this study, we use data collected from multiple researchers across four states across the southeastern U.S. to test competing hypotheses that describe disease impacts on the number of reproductive female bats with respect to WNS. Female bats enter hibernation with a set amount of stored fat and may have opportunities to resupply throughout winter if winters are mild [34]. During short and/or mild winters, females may leave hibernation with excess fat, allowing immediate fertilization. Alternatively, during long or severe winters, females may end hibernation with no fat stores and need to resupply, thus delaying, or even foregoing, fertilization and pregnancy. As WNS increases energetic stress during hibernation, we may see interactive effects of winter duration and severity with disease on the abundance of reproductive females. Thus we hypothesize that the number of reproductive females will be a function of duration and severity of winter and time since WNS introduction, wherein we will see an increase in reproduction (i.e., pregnant, and lactating females) following short, mild winters in years post detection of WNS. On the other hand, we propose the hypothesis that reproductive females rely on foraging during the periods of spring, summer, and fall to build up their fat reserves before hibernation. As a result, we anticipate that favorable climate conditions leading up to hibernation will correlate positively with the abundance of reproductive females in the subsequent year.

## Materials and methods

### Data collection

We compiled capture data over the years 1989–2020 from four southeastern states (GA, KY, NC, TN; Fig 1) from published surveys [28, 35, 36] and state and federal datasets (S1 Table). A total of 9,561 reproductive female bats from 9 species in 261 counties were collected from 1989–2020. Only those individuals that had clearly defined reproductive conditions were included in the analyses.

We defined WNS status in two ways in our analyses: first, before or after WNS occurrence in each county and second, the year since WNS was first observed in each county. We obtained the year that WNS was first documented in each county in each state from the US Fish and Wildlife Service (whitenosesyndrome.org). We then calculated the years since WNS for each county by counting the number of years since WNS was first documented. Since we lacked individual WNS status, the binning of bats to county-wide WNS status was unavoidable. Though it is likely that bats inhabiting WNS positive roosts may not be positive for the fungus, we must make the assumption that all bats captured in this analyses were exposed to the disease.

Prior to the onset of WNS, survey efforts across North America in the form of both mist-netting and acoustics remained relatively low [37–39]. Due to changes in research objectives, post-WNS efforts increased substantially [40, 41]. We calculated survey effort as the total number of net nights per county per year to include as a covariate; we used this metric to match our county-wide WNS designation.

Finally, we obtained winter duration, winter severity, and pre-hibernation climatic variables. We determined the mean, maximum, and minimum elevation for each county with

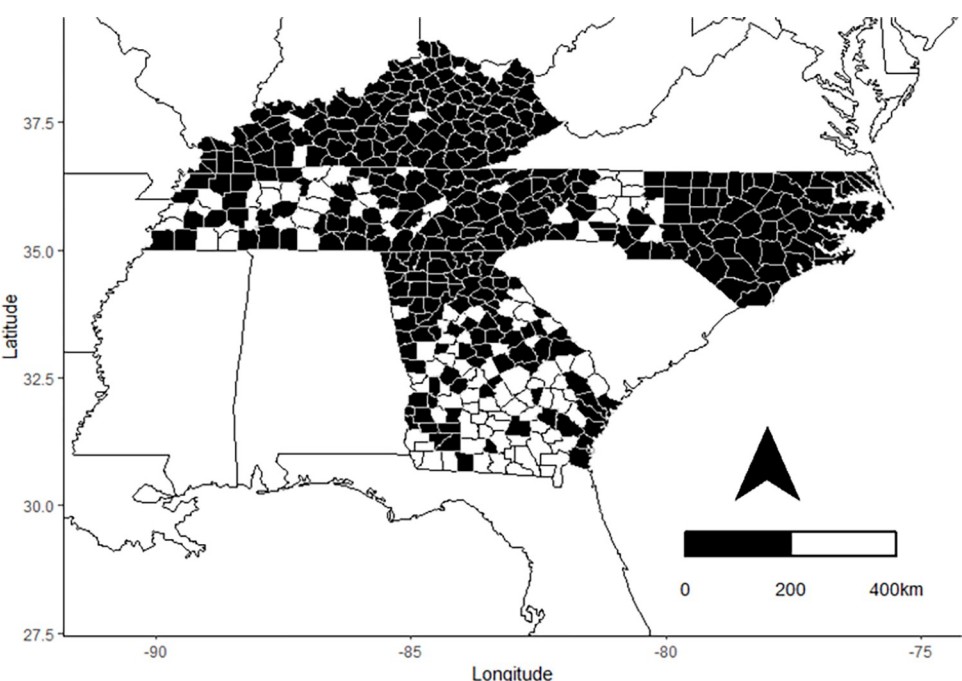

**Fig 1. Capture data.** Capture data from hibernating bat species were collected in 267 counties (black) in Tennessee, Kentucky, North Carolina, and Georgia, USA from 1989–2020.

capture data using a digital elevation model (DEM) provided by Wang et al [42]. We calculated a suite of summer, winter, spring, and autumn metrics (see S2 Table) for each county [43] with the ClimateNA v5.10 software package (http://tinyurl.com/ClimateNA), based on methodology described by Wang et al. [42]. Using methods described by Hranac et al. [44], we predicted the mean winter duration for each county based on county center latitude, elevation, and the number of days in frost gathered from ClimateNA.

We used a principal component analysis to find the key combinations of climate variables from pre-hibernation months per county per year that described most of the variation in the climate data (>90% [45]; S2 Table). The first two principal components had eigenvalues > 1 [46] and were used as covariates in our statistical models (PC1 = 1.75 and PC2 = 1.55; see S3 Table, S1 Fig). We interpreted the components by selecting the predictor variables with the highest eigenvectors (> |0.40|) associated with each component ([47] see S4 Table, S2 Fig). Independent variables for PC1 (noted as "humidity component") with the highest eigenvectors included mean summer relative humidity (|0.44|) and mean spring relative humidity (|0.46|; see S4 Table). Independent variables for PC2 (noted as "temperature component") with the highest eigenvectors included number of summer days above 18˚C (|0.43|) and mean annual temperature (|0.49|; see S2 Fig). The humidity component and temperature component retained 61% of the variances contained in the data (S4 Table).

## Statistical models

We developed a suite of generalized linear mixed models to test the competing hypotheses of the impacts of WNS, winter duration, and local climate on the abundance of reproductive female bats. In all models, the response variable was the total number of reproductive females (pregnant, lactating, or post-lactating) per WNS susceptibility group per county per year; we used the abundance of all reproductive females as our response rather than the abundance of

each reproductive class because of low sample sizes post-WNS. Species susceptibility to WNS was grouped based on criteria discussed in Jackson et al. (2022): low susceptibility (*E. fuscus*, *Myotis leibii*, *Myotis grisescens*), mild susceptibility (*Myotis sodalis*), and high susceptibility (*P. subflavus*, *M. lucifugus*, *Myotis septentrionalis*). We included year and county as random effects and sampling effort (log[capture nights per year]) as a fixed effect in all models. To deal with the non-normality of count data, we included a Poisson distribution with a log-link to transform the error distribution of the residuals.

Our explanatory variables varied depending on the hypothesis of question (Table 1). We included fixed effects of winter duration (predicted days in winter per county), the two principal components for pre-hibernation climate variables per county (humidity component and temperature component), and a combination of all three variables as they are not mutually exclusive (Table 1). We also incorporated the effects of WNS as a fixed effect in each model in two ways: first as presence/absence, with presence dictated by the year first observed in the county in which that bat was surveyed, and second, year(s) since WNS was reported, to allow for variation in the effect of disease over time. All continuous variables were centered on the mean and scaled by standard deviation.

We conducted model comparisons using the second-order Akaike's Information Criterion ($\Delta$AICc; Akaike 1973 [48]), giving preference to models with the lowest $\Delta$AICc values as the most plausible options. To address overdispersion, which is common in Poisson regression, we employed the $X^2$ approximation of the residual variance (Zuur et al. 2009 [49]). Models with a $c_t$ ratio < 1 were considered overdispersed. We compared models with a $\Delta$AICc within 2 units, utilizing model weights, and provided adjusted standard errors accordingly. The explanatory power of the fixed effects and the overall selected model was assessed through

**Table 1. Models from statistical analyses.** Candidate models predicting number of reproductive female bats (pregnant, lactating, post-lactating) of southeastern bat species with Akaike information criterion for over-dispersed data ($\Delta$AIC$_c$), log-likelihood values (*LL*), number of parameters (*K*), and AIC$_c$ weights (*w$_t$*). All models had a fixed effect of survey effort (log[capture nights per year]), WNS susceptibility grouping (low susceptibility: *Eptesicus fuscus*, *Myotis leibii*, *Myotis grisescens*; mild susceptibility: *Myotis sodalis*; and high susceptibility: *Perimyotis subflavus*, *Myotis lucifugus*, *Myotis septentrionalis*), and random effects of year and county.

| Model | $\Delta$AICc | LL | K | $w_t$ |
|---|---|---|---|---|
| Years with WNS * Pre-hibernation temperature (PC2) * Winter duration | 0.00 | -11012.97 | 26 | 0.999 |
| Years with WNS * Winter duration | 22.07 | -11036.07 | 14 | < 0.001 |
| Years with WNS * Pre-hibernation temperature (PC2) | 23.39 | -11036.73 | 14 | < 0.001 |
| Winter duration + Pre-hibernation humidity (PC2) | 34.93 | -11042.50 | 14 | < 0.001 |
| Pre/post-WNS * Pre-hibernation temperature (PC2) * Winter duration | 37.55 | -11031.74 | 26 | < 0.001 |
| Years with WNS | 39.97 | -11051.04 | 8 | < 0.001 |
| Years with WNS * Pre-hibernation humidity (PC1) * Winter duration | 40.94 | -11033.44 | 26 | < 0.001 |
| Years with WNS * Pre-hibernation humidity (PC1) | 49.29 | -11049.68 | 14 | < 0.001 |
| Pre/post-WNS * Winter duration | 51.62 | -11050.84 | 14 | < 0.001 |
| Winter duration | 55.36 | -11058.73 | 8 | < 0.001 |
| Pre-hibernation temperature (PC2) | 58.66 | -11060.38 | 8 | < 0.001 |
| Pre-hibernation humidity (PC1) and temperature (PC2) | 59.37 | -11057.73 | 11 | < 0.001 |
| Pre/post-WNS * Pre-hibernation temperature (PC2) | 59.83 | -11054.95 | 14 | < 0.001 |
| Winter duration + Pre-hibernation humidity (PC1) | 60.34 | -11055.21 | 14 | < 0.001 |
| Pre/post-WNS | 63.26 | -11062.68 | 8 | < 0.001 |
| Pre/post-WNS * Pre-hibernation humidity (PC1) * Winter duration | 63.82 | -11044.88 | 26 | < 0.001 |
| WNS susceptibility group only | 65.75 | -11066.93 | 5 | < 0.001 |
| Pre/post-WNS * Pre-hibernation humidity (PC1) | 67.26 | -11058.66 | 14 | < 0.001 |
| Pre-hibernation humidity (PC1) | 68.07 | -11065.09 | 8 | < 0.001 |

calculated R$^2$ values. The significance of individual variables in the top-ranked model was determined using p-values (α = 0.05). All statistical analyses were performed using R v3.3.3 [50], with the R package *lme4* [51].

## Results

The dataset contained 31 years of capture data from 234 counties in the four states (Figs 1 and 2). There were a total of 11,896 reproductive females (pregnant = 2,069, lactating = 2,713, post-lactating = 2,601) across all species groups (low susceptibility = 3,164, mild susceptibility = 1,127, high susceptibility = 3,062). Of the low susceptible species, there were many more *E. fuscus* (n = 2,648) then *M. leibii* (n = 212) and *M. grisescens* (n = 334). Of the highly susceptible species, there were more *M. septentrionalis* (n = 1,490) then *P. subflavus* (n = 943) or *M. lucifugus* (n = 629). Finally, there were 1,127 *M. sodalis* in the mildly susceptible group.

The strongest support for the abundance of reproductive females was observed in the interaction among species WNS susceptibility, winter duration, pre-hibernation climate (PC 2 = temperature), and the time elapsed since the initial documentation of the disease (AIC$_C$ = 22078.13, LL = -11012.97, K = 26, w$_t$ = 0.99); this model exhibited a difference > 20 AICc compared to the subsequent top-performing model (Table 1). Species exhibiting mild susceptibility and those highly susceptible to WNS experienced a decline in the abundance of reproductive females with the emergence of WNS, as expected (mild susceptibility: (β = -0.03 ± 0.02 SE, p = 0.04; high susceptibility: (β = -0.04 ± 0.01 SE, p < 0.0001; Table 2, Fig 2B).

The relationship between the abundance of reproductive females and winter duration displayed a negative correlation (β = -0.10 ± 0.02 SE, p < 0.0001; Fig 3A), irrespective of the species' susceptibility to WNS (p > 0.05 for all WNS susceptibility species groups) or the time elapsed since the onset of WNS (p = 0.06). Conversely, the impact of pre-hibernation climate, specifically temperature (PC 2), demonstrated a positive association with female abundance (β = 0.05 ± 0.02 SE, p = 0.001; Fig 3B). When considering the interaction between WNS susceptibility groups and climate, only the mildly susceptible species (*M. sodalis*) exhibited a distinct relationship compared to the other groups (β = -0.11 ± 0.03 SE, p < 0.001). The influence of pre-hibernation climate did not change based on the onset of WNS (p = 0.33) and did not mitigate the effect of winter duration on female abundance (p = 0.91). Among species highly susceptible to WNS (*P. subflavus*, *M. lucifugus*, *M. septentrionalis*), their population decreased with longer winters, even when accounting for the interaction with pre-hibernation temperature (β = -0.04 ± 0.02 SE, p = 0.02). In contrast, mildly susceptible species exhibited an increase in population with warmer pre-hibernation climates, even in the presence of WNS (β = 0.04 ± 0.01 SE, p = 0.004).

## Discussion

Our analysis of a long-term historical dataset provides evidence highlighting the substantial impact of pre-hibernation climate and winter duration on the population of reproductive females in species susceptible to white-nose syndrome (WNS) upon its onset. In particular, our findings demonstrate that warmer pre-hibernation climate conditions are associated with an increase in the number of reproductive females, while longer winters correlate with a decrease. These results coincide with previous work that has noted the importance of forage resources for survival [52–54]. Following the appearance of WNS, there was a notable reduction in reproductive females in those species susceptible to the disease, with the decline being more prominent among those classified as highly susceptible (*P. subflavus*, *M. lucifugus*, *M. septentrionalis*). These outcomes are in line with our expectations, given the predictable influence of pre-hibernation climate, winter duration, and WNS on bat energetics.

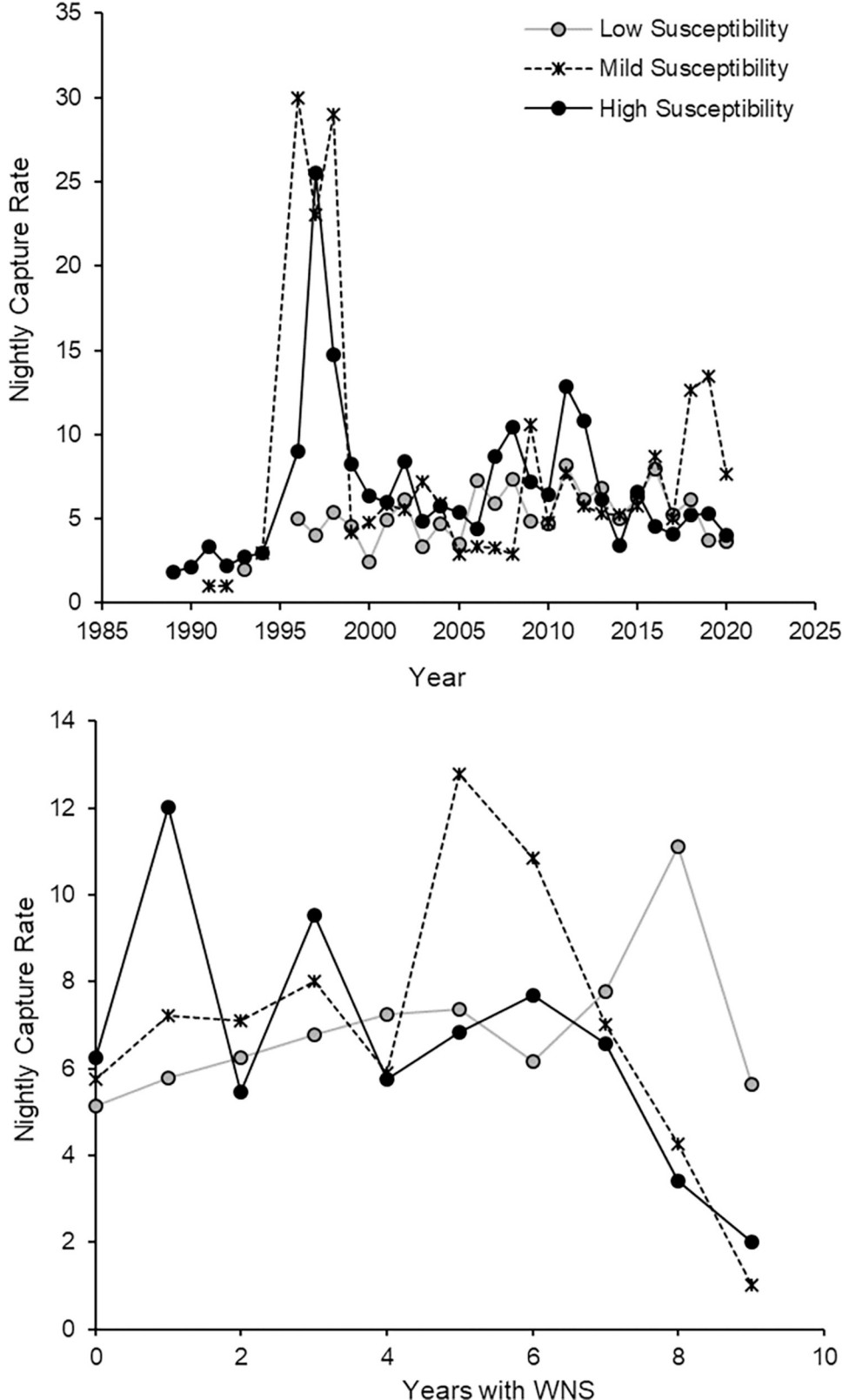

**Fig 2. Mean nightly capture rates.** Mean nightly capture rates of reproductive (pregnant, lactating, post-lactating) females for three white-nose syndrome susceptibility groups (low susceptibility: *Eptesicus fuscus*, *Myotis leibii*, *Myotis*

*grisescens*; mild susceptibility: *Myotis sodalis*; and high susceptibility: *Perimyotis subflavus*, *Myotis lucifugus*, *Myotis septentrionalis*) summarized by county and state for A) each year from 1989–2020 and B) since the onset of WNS in the southeastern United States.

Environmental factors highly correlated with the number of reproductive females included the number of days exceeding 18°C during summer and the mean annual temperature (PC 2; refer to S4 Table). Not surprisingly, this aligns with existing evidence indicating that bats in temperate zones heavily rely on favorable foraging conditions before entering hibernation. Elevated temperatures in summer and spring, along with increased relative humidity, directly impact the availability of prey [52, 55]. When prey is abundant prior to hibernation and weather conditions permit successful foraging endeavors, the likelihood of bats accumulating fat reserves rises. This heightened fat storage enhances their potential to endure challenging winters and the pressures of WNS [19, 44]. Furthermore, complementary research proposes a link between the accumulation of fat before hibernation and the probability of successful

**Table 2. Summary statistics for top model.** Summary statistics for top model predicting number of reproductive female bats (pregnant, lactating, post-lactating) per county per year of southeastern bat species (low WNS susceptibility: *Eptesicus fuscus*, *Myotis leibii*, *Myotis grisescens*; mild WNS susceptibility: *Myotis sodalis*; and high WNS susceptibility: *Perimyotis subflavus*, *Myotis lucifugus*, *Myotis septentrionalis*) with coefficient estimate (β), standard error (SE), and p-value. WNS susceptibility variables are against the reference of low susceptible species. The temperature variable is the composite principal component 2 of pre-hibernation climate variables; PC 2 included the number of days exceeding 18°C during summer and the mean annual temperature. Years since WNS is the number of years since WNS was first observed in the county.

| Model Variable | β | SE | p-value |
|---|---|---|---|
| Intercept | 0.38 | 0.03 | *< 0.0001* |
| High susceptibility | 0.02 | 0.03 | 0.55 |
| Mild susceptibility | 0.09 | 0.04 | *0.02* |
| Years since WNS | 0.04 | 0.01 | *< 0.0001* |
| Winter duration | -0.10 | 0.02 | *< 0.0001* |
| Temperature | 0.05 | 0.02 | *< 0.0001* |
| log(number of capture nights) | -0.03 | 0.01 | 0.06 |
| High susceptibility * Years since WNS | -0.04 | 0.01 | *< 0.001* |
| Mild susceptibility * Years since WNS | -0.03 | 0.02 | *0.04* |
| High susceptibility * Winter duration | -0.02 | 0.03 | 0.63 |
| Mild susceptibility * Winter duration | 0.08 | 0.05 | 0.15 |
| Winter duration * Years since WNS | 0.02 | 0.01 | 0.06 |
| High susceptibility * Temperature | 0.01 | 0.02 | 0.43 |
| Mild susceptibility * Temperature | -0.11 | 0.03 | *< 0.0001* |
| Temperature * Years since WNS | -0.01 | 0.01 | 0.33 |
| Winter duration * Temperature | 0.00 | 0.01 | 0.91 |
| High susceptibility * Years since WNS * Winter duration | 0.01 | 0.01 | 0.44 |
| Mild susceptibility * Years since WNS * Winter duration | 0.03 | 0.02 | 0.12 |
| High susceptibility * Years since WNS * Temperature | 0.00 | 0.01 | 0.58 |
| Mild susceptibility * Years since WNS * Temperature | 0.04 | 0.01 | *< 0.0001* |
| High susceptibility * Winter duration * Temperature | -0.04 | 0.02 | *0.02* |
| Mild susceptibility * Winter duration * Temperature | 0.04 | 0.03 | 0.16 |
| Years since WNS * Temperature * Winter duration | -0.01 | 0.01 | 0.28 |
| High susceptibility * Years since WNS * Temperature * Winter duration | 0.01 | 0.01 | 0.26 |
| Mild susceptibility * Years since WNS * Temperature * Winter duration | 0.00 | 0.02 | 0.76 |

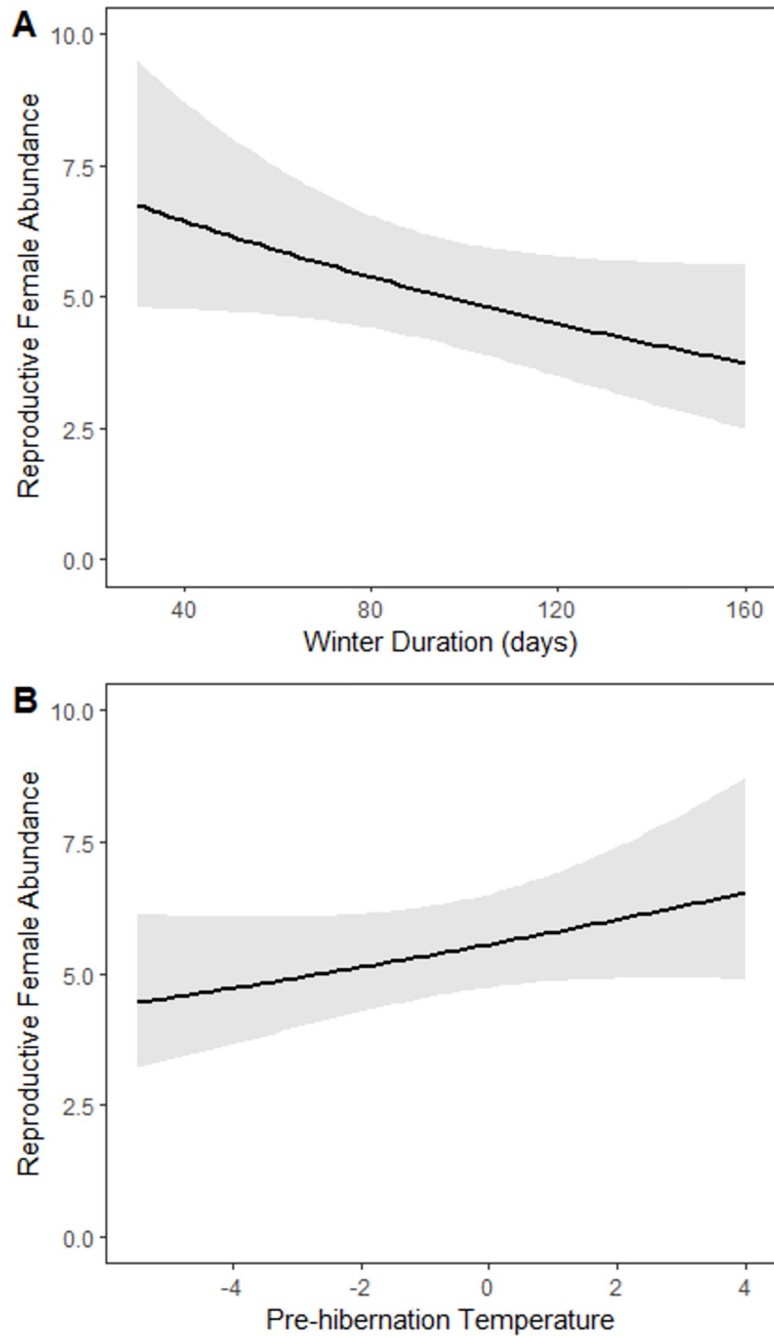

**Fig 3. Predicted number of females from analyses.** Predicted number of reproductive females of seven southeastern bat species against A) mean winter duration (in days) and B) the principal component (PC 2) that represents pre-hibernation temperature. Both predictive models set all other variables (survey effort (log[capture nights per year], time of WNS exposure) to the mean. Error bands represent 95% confidence intervals.

reproduction [21], a notion supported by our data. Nonetheless, the connection between pre-hibernation body condition and successful reproduction has not been exclusively tested.

In addition to pre-hibernation climate conditions conducive to ample fat storage, our study also demonstrates a correlation between the number of reproductive females and the duration of winter in species susceptible to WNS. Notably, the number of reproductive females in

WNS-susceptible species was lower following lengthier winters. This relationship is to be expected, given that extended winters demand greater fat reserves for prolonged hibernation. Consequently, bats affected by WNS appear to fare better in regions characterized by shorter winter durations [44]. Similarly, surviving hibernation with the presence of WNS does not guarantee that a bat will possess the necessary energy for successful offspring rearing. Johnson et al. [56] presented evidence indicating that the energetic demands on WNS-affected bats intensify as they endeavor to recover and engage in foraging activities to replenish the fat stores lost upon emerging from hibernation. This insight suggests that females might opt to forego reproduction when confronted with the dual stressors of prolonged winter durations and the impacts of WNS, aligning with the observations in our findings.

Our results add to the growing literature that there are there are energetic costs for reproductive-age females contending with the effects of WNS [38, 57]. We observed a post-WNS decline in captures of reproductive females of WNS-susceptible species (*M. sodalis*, *M. lucifugus*, *M. septentrionalis*, and *P. subflavus*). Pettit and O'Keefe [58] also observed negative effects of WNS on reproduction and hypothesized that WNS-induced fat depletion could force female bats to miscarry pups or prolong the development of young. Lengthening the gestation period after hibernation could negatively impact pup development prior to hibernation. If pups are born later in the summer, they might not have enough time to build up sufficient fat stores before going into their first hibernation, thus decreasing their probability of survival over winter, thereby further exacerbating population declines from WNS. Further evidence that WNS may be delaying reproduction is needed to determine the relevance of this phenomenon. We suggest future studies on the mechanisms of WNS that can lead to pregnancy loss.

Our results suggest that ecological release may be occurring with species less susceptible to WNS (*E. fuscus*, *Myotis leibii*, *M. grisescens*), allowing them to use resources that may have otherwise been depleted by other species [59]. For example, Jachowski et al. [60] found that following dramatic declines of the once common *M. lucifugus*, species not susceptible to WNS increased their spatial and temporal overlap with *M. lucifugus* in northwestern New York. Their results showed that due to reduced interspecific competition, some species were able to occupy aerial foraging space once used by these WNS-impacted species. Future investigations should focus on how these potential changes in niche partitioning and decreased competition between susceptible and non-susceptible species result in long-term changes to bat community structure.

Finally, the results of the methods and findings of this work would lend themselves to an increased, perhaps country wide study using archival data and data generated by ongoing surveillance efforts [such as NABat; 61]. It would also be telling to incorporate these results into demographic models of WNS susceptible bat populations to study how demographic shifts may contribute to population crashes. Most research on how bats survive hibernation to successfully reproduce has either focused on the impacts of winter duration or local climate or disease, but not the combined effects of all three. We conclude that WNS, pre-hibernation climate, and winter duration all impact reproductive success for southeastern bat species. Using data spanning from 1989–2020, we were able to assess the effects of disease and how reproductive success may vary with local climate. Predicting the interplay among these three variables must consider the variation of disease impacts across both susceptible and non-susceptible species across different parts of their range, and how local variation can modulate reproduction. Our results highly the key components important to reproductive success in southeastern bat species and provide a necessary stepping stone to further our understanding of the costs associated with surviving different life history stages in the context of disease.

## Supporting information

**S1 Fig. Eigenvalues from principal component analysis.** Eigenvalues and the percent variance explained by each principal component from a principal component analysis summarizing climate variables (mean annual temperature, number of summer days above 18˚C, number of spring days above 18˚C, mean annual precipitation, spring mean relative humidity, summer mean relative humidity, autumn mean relative humidity, number of frost-free days, and number of spring days below 0˚C) from Tennessee, North Carolina, Georgia, and Kentucky from 1989–2020.
(DOCX)

**S2 Fig. Factor loadings from principal component analysis.** Factor loadings for the first two principal components of a principal component analysis summarizing climate variables for Tennessee, North Carolina, Georgia, and Kentucky from 1989–2020. Identifiers of the variables: mean annual temperature, number of summer days above 18˚C, number of spring days above 18˚C, mean annual precipitation, spring mean relative humidity, summer mean relative humidity, autumn mean relative humidity, number of frost-free days, and number of spring days below 0˚C.
(DOCX)

**S1 Table. Capture data used in the analyses.** Capture data from hibernating bat species were collected in 267 counties (black) in Tennessee, Kentucky, North Carolina, and Georgia, USA from 1989–2020. We report only the capture data available in published articles; other data available from listed state and federal agencies (see Data Availability statement).
(DOCX)

**S2 Table. Climate variables used in principal component analysis.** Winter (December [previous year for an individual year], January, and February), spring (March, April, and May), summer (June, July, and August), autumn (September, October, and November), and annual severity metrics calculated per county for Tennessee, North Carolina, Georgia, and Kentucky for each year from 1989–2020. These metrics were used in a principal component analysis to describe pre-hibernation climate variables.
(DOCX)

**S3 Table. Eigenvalues from principal component analysis.** Eigenvalues and proportion of total variance explained by each axis derived from a principal component analysis of pre-hibernation climate data for Tennessee, North Carolina, Georgia, and Kentucky from 1989–2020.
(DOCX)

**S4 Table. Eigenvectors from principal component analysis.** Eigenvectors associated with pre-hibernation climate variables used in a principal component analysis summarizing climate metrics for Tennessee, North Carolina, Georgia, and Kentucky from 1989–2020.
(DOCX)

## Acknowledgments

We would like to thank T. Walker, B. Williams, and N. Deans for help with data collection, C. R. Hranac for winter duration model development and E. Rehm and C. Gienger for comments on the manuscript. We would like to acknowledge R. Bernard, Fort Campbell Fish & Wildlife, Kentucky Fish & Wildlife, Georgia Department of Natural Resources, North Carolina Wildlife Resources Commission, and Tennessee Wildlife Resources Agency for data.

## Author Contributions

**Conceptualization:** Sarah K. Krueger, Catherine G. Haase.

**Data curation:** Sarah K. Krueger, Sarah C. Williams, Joy M. O'Keefe, Gene A. Zirkle, Catherine G. Haase.

**Formal analysis:** Sarah K. Krueger.

**Funding acquisition:** Catherine G. Haase.

**Investigation:** Sarah K. Krueger, Sarah C. Williams, Joy M. O'Keefe, Gene A. Zirkle, Catherine G. Haase.

**Methodology:** Sarah K. Krueger, Catherine G. Haase.

**Project administration:** Sarah C. Williams, Gene A. Zirkle.

**Validation:** Sarah K. Krueger.

**Visualization:** Sarah K. Krueger.

**Writing – original draft:** Sarah K. Krueger.

**Writing – review & editing:** Sarah K. Krueger, Joy M. O'Keefe, Catherine G. Haase.

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
