## [Decision Letter · Decision Letter 0]

31 Oct 2023

PONE-D-23-27009White-nose syndrome, winter duration, and pre-hibernation climate impact abundance of reproductive female batsPLOS ONE

Dear Dr. Krueger,

Thank you for submitting your manuscript to PLOS ONE. After careful consideration, we feel that it has merit but does not fully meet PLOS ONE’s publication criteria as it currently stands. Therefore, we invite you to submit a revised version of the manuscript that addresses the points raised during the review process. In addition to the very minor comments made by the two reviewers, please note that all data necessary to replicate your study should be publicly available without restriction. Your data accessibility statement mentions datasets that cannot be published and/or that are already available.- Please provide details about the ratio between these two types of datasets.- For data that cannot be made available publicly: please provide the reason why it cannot be shared and the contact information others would need to apply to gain access to the data.- For previously published datasets: please note that Rojas et al. 2017 and O'Keefe et al. 2019 are not open access papers. The data is therefore not publicly available. I'd suggest to provide a table merging the datasets that were included in your analysis. I guess that you may already have compiled such a table for the statistical analyses. You may submit it as a supplementary table in your revised manuscript and add an extra row mentioning the data source (Grieneisen, Rojas et al. 2017 or O'Keefe et al. 2019).

We look forward to receiving your revised manuscript.

Kind regards,

Camille Lebarbenchon

Academic Editor

PLOS ONE

Journal Requirements:

"Funding for this research and support for S. Krueger was provided by an Intergovernmental Support Agreement with the U.S. Army (Order # CAMP-IGSA-20-02)."

4. We note that you have referenced (unpublished) on page 6, which has currently not yet been accepted for publication. Please remove this from your References and amend this to state in the body of your manuscript: (ie “Bewick et al. [Unpublished]”) as detailed online in our guide for authors

6. We note that [Figure 1] in your submission contain [map/satellite] images which may be copyrighted. All PLOS content is published under the Creative Commons Attribution License (CC BY 4.0), which means that the manuscript, images, and Supporting Information files will be freely available online, and any third party is permitted to access, download, copy, distribute, and use these materials in any way, even commercially, with proper attribution. For these reasons, we cannot publish previously copyrighted maps or satellite images created using proprietary data, such as Google software (Google Maps, Street View, and Earth). For more information, see our copyright guidelines: http://journals.plos.org/plosone/s/licenses-and-copyright.

Reviewers' comments:

Reviewer's Responses to Questions

**Comments to the Author**

1. Is the manuscript technically sound, and do the data support the conclusions?

Reviewer #1: Yes

Reviewer #2: Yes

2. Has the statistical analysis been performed appropriately and rigorously? 

Reviewer #1: Yes

Reviewer #2: Yes

3. Have the authors made all data underlying the findings in their manuscript fully available?

Reviewer #1: Yes

Reviewer #2: Yes

4. Is the manuscript presented in an intelligible fashion and written in standard English?

Reviewer #1: Yes

Reviewer #2: Yes

5. Review Comments to the Author

Reviewer #1: I happened to be a reviewer for a previous version of this submitted to a different journal (subsequently rejected) and was very impressed to see the updates and changes the authors incorporated into this version. The writing is much improved, therefore the storytelling flows nicely. The inclusion of varying susceptibility and analyzing the changes in female population/reproduction based on those classes makes so much more sense and is a story that is easily tractable. Great job. I only have minor comments throughout the manuscript, jotted down as comments in the PDF.

Reviewer #2: A really well written and informative manuscript. It was refreshing to review a manuscript where I wasn't having to make lots of comments regarding writing, analysis, etc. I had very few comments and they were mostly minor typos. I have highlighted these in the attached document. Again, great job! I look forward to seeing this in press

6. PLOS authors have the option to publish the peer review history of their article (what does this mean?). If published, this will include your full peer review and any attached files.

Reviewer #1: No

Reviewer #2: No

---

## [Author Response · Author response to Decision Letter 0]

20 Dec 2023

We would like to thank the reviewers for their feedback regarding our manuscript. Our responses to their comments are as followed:

All editorial comments (grammar, punctuation, etc.) were changed as suggested.

Reviewer 1:

• Added line numbers.

• Line 110: Removed “colder, drier conditions”

• Line 25: Removed “male”

• Line 28: We agree with the statement and were unsure what the reviewers were referring to.

• Line 29: We included another mammal example in the first paragraph, and thus feel that we do not need to include another example but rather focus on bats.

• Line 40: We removed the non-sequitur. 

• Line 45: Added citation.

• Line 47: Removed common name

• Line 53: Removed “devastating” 

• Line 59: We reworded to be correct

• Line 63: Added suggested citation

• Line 64: Rewrote as suggested

• Line 77: Added text about hibernation behavior as suggested.

• Like 79: Rewrote as suggested

• Line 80: Rewrote as suggested

• Line 82: Removed “in bat populations” as suggested

• Lines 83-84: Rewrote and added suggested citation

• Lines 94-96: We do not account for mid-winter foraging activities but are assuming none in these areas. Though we know that assumption is more than likely false, because we are using data from a variety of sources from across multiple years, we do not have the data we need to include that component. However, we can attribute warmer conditions to allow for these foraging activities, and thus this question could inherently be answered by the impact of climate. We are just unable to untangle that part in this work.

• Line 116: Added clarification

• Line 119: We removed the confusing text to clarify using county-level summaries.

• Line 180: Clarified sentence

• Line 241: Added citations as suggested

• Line 294: Added NABat

Reviewer 2

• Line 7: We are not sure what the reviewer was referring to as we have the range of susceptibility already in that sentence

• Line 114: Changed the error

• Line 119: We removed that statement

• Line 184: Changed word as suggested

• Line 235: Changed as suggested

---

## [Editor Report · Decision Letter 1]

26 Jan 2024

White-nose syndrome, winter duration, and pre-hibernation climate impact abundance of reproductive female bats

PONE-D-23-27009R1

Dear Dr. Krueger,

We’re pleased to inform you that your manuscript has been judged scientifically suitable for publication and will be formally accepted for publication once it meets all outstanding technical requirements.

Kind regards,

Camille Lebarbenchon

Academic Editor

PLOS ONE
---

## [Editor Report · Acceptance letter]

26 Mar 2024

PONE-D-23-27009R1 

PLOS ONE

Dear Dr. Krueger, 

I'm pleased to inform you that your manuscript has been deemed suitable for publication in PLOS ONE. Congratulations! Your manuscript is now being handed over to our production team.

Kind regards, 

on behalf of

Dr. Camille Lebarbenchon 

Academic Editor

PLOS ONE